# More than Just Bread and Wine: Using Yeast to Understand Inherited Cytochrome Oxidase Deficiencies in Humans

**DOI:** 10.3390/ijms25073814

**Published:** 2024-03-29

**Authors:** Chenelle A. Caron-Godon, Emma Collington, Jessica L. Wolf, Genna Coletta, D. Moira Glerum

**Affiliations:** 1Department of Biology, University of Waterloo, Waterloo, ON N2L 3G1, Canada; cacarongodon@uwaterloo.ca (C.A.C.-G.); e4collington@uwaterloo.ca (E.C.); jessica.wolf@mail.mcgill.ca (J.L.W.); gcoletta@uwaterloo.ca (G.C.); 2Waterloo Institute for Nanotechnology, University of Waterloo, Waterloo, ON N2L 3G1, Canada

**Keywords:** mitochondrial disease, yeast model, COX assembly, copper transfer, heme A biosynthesis

## Abstract

Inherited defects in cytochrome *c* oxidase (COX) are associated with a substantial subset of diseases adversely affecting the structure and function of the mitochondrial respiratory chain. This multi-subunit enzyme consists of 14 subunits and numerous cofactors, and it requires the function of some 30 proteins to assemble. COX assembly was first shown to be the primary defect in the majority of COX deficiencies 36 years ago. Over the last three decades, most COX assembly genes have been identified in the yeast *Saccharomyces cerevisiae*, and studies in yeast have proven instrumental in testing the impact of mutations identified in patients with a specific COX deficiency. The advent of accessible genome-wide sequencing capabilities has led to more patient mutations being identified, with the subsequent identification of several new COX assembly factors. However, the lack of genotype–phenotype correlations and the large number of genes involved in generating a functional COX mean that functional studies must be undertaken to assign a genetic variant as being causal. In this review, we provide a brief overview of the use of yeast as a model system and briefly compare the COX assembly process in yeast and humans. We focus primarily on the studies in yeast that have allowed us to both identify new COX assembly factors and to demonstrate the pathogenicity of a subset of the mutations that have been identified in patients with inherited defects in COX. We conclude with an overview of the areas in which studies in yeast are likely to continue to contribute to progress in understanding disease arising from inherited COX deficiencies.

## 1. Introduction

The generation of a functional mitochondrion requires the input of both the nuclear genome and mitochondrial DNA (mtDNA) to generate the thousand-plus proteins that must find their way to one of several mitochondrial destinations: two different membranes and three different submitochondrial spaces. Not surprisingly, therefore, inherited diseases affecting mitochondria, the primary producers of cellular energy, result in a bewildering variety of different clinical phenotypes. Diseases of mitochondrial dysfunction have been reported for more than four decades now, with a large number affecting the function of the respiratory chain; a significant subset of these are characterized by specific deficiencies associated with cytochrome *c* oxidase (COX) [1]. COX is unique with regard to the large number of proteins required for assembly of the holoenzyme, with numerous (~30) proteins required to support synthesis and membrane insertion of the core subunits, as well as providing the requisite copper atoms and heme A molecules [1]. The complex genetics (i.e., contributions from both the mitochondrial and nuclear genomes) and large number of genes required to form an active COX have made identifying and characterizing pathologies arising from defective COX assembly a challenging task. While advances in sequencing technologies have vastly improved the ability to diagnose/identify genetic defects associated with cases of COX deficiency, determining whether a molecular variant is causative for disease can still greatly benefit from the use of a tractable model system.

### Yeast as a Model System for Human Cell Biology

The yeast *Saccharomyces cerevisiae* is a commonly used model in cell biology for a variety of reasons. Since the 1980s, there have been well-established approaches for genetic manipulation, which, combined with a relatively inexpensive means of propagation, have facilitated many advances in understanding human cell biology and enabled yeast to become the eukaryotic single-celled workhorse of biotechnology. Equally important is the fact that cellular processes such as endoplasmic reticulum-associated protein degradation, heat shock, chaperone functions, autophagy, and protein translation, folding, and secretion are all highly conserved between yeast and humans [2]. This high degree of conservation also extends to signal transduction processes and implies that signal cross-talk, regulation hierarchies, and protein–protein interactions are similar in these two evolutionarily distant organisms [2]. One of the most profound examples of harnessing the power of yeast genetics to improve our understanding of human disease is provided by the identification and characterization of cell cycle proteins, which have directly informed our understanding and further study of cancer in humans and for which Leland Hartwell and Paul Nurse were awarded the Nobel Prize in Physiology or Medicine in 2001. Indeed, 47% of yeast genes that have a single human orthologue and have been shown to be essential have been successfully replaced by their human orthologue [3]. The high degree of similarity between yeast and human genes, along with similarities in cellular processes, has rendered yeast an incredibly powerful model for elucidating basic tenets of cell biology and allows it to remain an indispensable model for understanding human disease. Even when yeasts do not share a close orthologue for a protein present in mammalian cells, it is often feasible to create a humanized homologue of the gene to study in yeast [4]. Indeed, expression of mammalian disease-causing genes with a yeast orthologue often results in complementation of the loss-of-function phenotype [5].

In the 1980s, a number of yeast respiratory mutant collections [6,7] were generated, and these have served as the foundation for the incredibly fruitful identification of proteins required for mitochondrial biogenesis and metabolism. The mitochondria in yeast are remarkably similar to the human organelles, in both structure and function; human COX activity assays are frequently carried out using yeast cytochrome *c*, while yeast COX activity assays routinely use mammalian cytochrome *c*. In the context of studying COX assembly and related defects, *Saccharomyces cerevisiae* has a few distinct advantages that have made it a preferred model. First and foremost, *S. cerevisiae* is a facultative anaerobe, meaning the organism can grow on both fermentable and non-fermentable carbon sources. When a yeast strain is rendered respiration deficient by a mutation, growth is supported on fermentable carbon sources, such as glucose or galactose [8]. As in humans, *Saccharomyces cerevisiae* relies solely on COX for oxidative respiration, lacking the alternative ubiquinol oxidase that is found in some species of yeasts and other eukaryotic organisms [9,10]. Furthermore, *S. cerevisiae* is a well-suited model due to the high degree of similarity between the COX assembly processes in mammalian and yeast cells [11]. For human disease research in particular, *S. cerevisiae* allows compound heterozygous mutations to be studied separately or together, given the ability of yeast to exist in either the haploid or diploid state. Studies in yeast have contributed more than any other model organism to our understanding of COX assembly and the genes that are implicated in diseases arising from assembly defects. In this review, we highlight the inherited human COX deficiencies for which either prior or subsequent modeling in yeast has provided a deeper understanding of the disease phenotype in patients.

## 2. The COX Assembly Pathways in Humans and Yeast

The assembly of a functional COX complex requires the carefully coordinated action of at least 30 proteins, in addition to the 14 subunit constituents [12]. The exact roles for many of these subunits remain unclear, and investigations into the mechanisms behind the assembly of the holoenzyme are ongoing. The catalytic core of the complex is made up of the three mitochondrial-encoded subunits—COX1, COX2, COX3. Each of these are assembled into the mitochondrial inner membrane with the assistance of their distinct sets of assembly factors that stabilize the assembling apoenzyme in the membrane and ensure the appropriate insertion of essential cofactors [12]. Early research into COX assembly described the process as being linear, with subunits being added onto COX1 one after another [13]. The linear assembly model, which was supported by results that demonstrated that COX1 acted as a seed to which other subunits could join, has been largely replaced by the concept of modular COX assembly [14,15], wherein each of the three catalytic subunits is formed separately with the assistance of its own dedicated set of assembly factors [16]. These modules are then added to a seed module of nuclear assembly factors in a linear manner [12], with the distinct steps of the process referred to as S1–S4. S1 involves the formation of the COX1 module, which then joins the nuclear seed module in S2. During the S3 stage, the COX2 module, followed by the COX3 module, are added. With the addition of the final auxiliary (i.e., non-catalytic core) subunits (S4), COX assembly is complete [17]. Many of the assembly processes are shared between yeast and humans, although human cells have additional subunits and control mechanisms relative to yeast [17]. The main steps are thought to occur in a similar manner in yeast and humans and are depicted in a schematic format in Figure 1, with the proteins relevant to this review highlighted in bold. Since a detailed description of the COX assembly pathway(s) is well beyond the scope of this review, the reader is referred to several recent in-depth reviews on the subject [18,19,20].

In the context of human disease, it is intriguing that most COX deficiency-associated mutations have been identified in the nuclear genes encoding COX assembly factors [21], which is likely because loss of a COX subunit results in fatality during intra-uterine development. Reduced or defective COX assembly was first identified more than three decades ago as a cause of COX deficiencies [22], and, in the intervening years, more than half of the known COX assembly proteins have been found to be defective in cases of human mitochondrial disease. Indeed, mutations in genes encoding assembly factors were identified before mutations in the nuclear genes encoding COX subunits. For a comprehensive discussion of inherited COX deficiencies, the reader is directed to the excellent recent review by Brischigliaro and Zeviani [1]. In this review, we focus on the contributions of yeast studies to our understanding of human COX deficiencies, since studies in *Saccharomyces cerevisiae* have been used to identify and characterize many of the currently known COX assembly factors.

## 3. Defects Affecting Synthesis and Assembly of COX1

Much of our current understanding of COX assembly in humans has arisen through the combination of studies in yeast and various human cell types. Because the nomenclature of the genes and their encoded products were assigned in a ‘non-linear’ fashion, Table 1 provides an overview of the proteins involved in COX assembly in yeast and humans that are discussed in this review. In accordance with nomenclature conventions, human and yeast gene names are italicized and capitalized (*COX10*), while yeast mutant strains are italicized in lower case (*cox10*). We have chosen to capitalize yeast protein names, (COX10) as is the case for human proteins, although the reader will see that the convention in the older yeast literature uses Cox10p or Cox10 for protein names.

COX assembly in humans begins with the COX1 module during the S1 stage, in which the mtDNA-encoded *COX1* mRNA is stabilized by LRPPRC (leucine-rich pentatricopeptide repeat containing) [23], a function that is carried out by the homologous PET309 in yeast [24,25]. The translation of COX1 is stimulated by the nuclear-encoded protein TACO1 (transcriptional activator for cytochrome oxidase; DPC29 is the yeast homolog) [26]. Translational regulation of human *COX1* expression also involves the MITRAC (mitochondrial translation regulation assembly intermediate of COX) complex [27], which includes the assembly factors C12ORF62 (COX14 in yeast) [28,29] and MITRAC12 (COA3 in yeast) [30]. Together with a number of other proteins not yet found to be involved in human disease, these assembly factors form a dynamic complex with OXA1L in the mitochondrial inner membrane [27,31,32]. Interestingly, early COX assembly in humans also requires the formation of a seed module made up of nuclear-encoded subunits COX4 and COX5A, which interact with the COX1 module via C12ORF62 [33].

### 3.1. Defects Associated with COX1 Expression

#### 3.1.1. LRPPRC/PET309

In 2003, in a tour-de-force of integrative genomics, Mootha et al. [23] showed that mutations in *LRPPRC* underlay the Saguenay-Lac St. Jean form of Leigh syndrome (Leigh Syndrome, French Canadian; LSFC) [34], which results in a COX deficiency due to impaired assembly. LRPPRC had originally been identified on the basis of an affinity for lectins, suggesting it might be a carbohydrate-binding protein, which would not immediately be suggestive of mitochondrial involvement. However, the innovative genomics-based approach used by Mootha and colleagues was supported by their subsequent identification of homozygous A354V mutations in the French-Canadian patient cohort, which further supported the founder effect identified previously [34]. Interestingly, contemporaneous studies with LRPPRC (also referred to as LRP130) suggested the protein localized to both the nucleus and mitochondria and bound to mRNAs of both nuclear and mitochondrial origin [35], with subsequent work demonstrating that LRPPRC also interacts with other transcripts, including the *COX3* mRNA [36]. The identification of different homozygous and compound heterozygous mutations in non-French-Canadian Leigh Syndrome patients demonstrated the relevance of *LRPPRC* mutations to patients with COX deficiencies and further broadened the potential impact of these mutations by documenting an associated Complex I deficiency as well [37].

During their initial investigations, Mootha et al. identified a weakly homologous yeast protein, PET309, which was first identified in yeast as an integral inner mitochondrial membrane protein responsible either for stabilizing primary transcripts of *COX1* or in initiating their translation [24,38]. Given that biochemical analyses in yeast demonstrated a physical interaction between *PET309* and *COX1* transcripts [25], with a direct role for the PPR motifs in that activity, the suggestion that PET309 and LRPPRC are not true orthologues [39] does not appear to hold true. The ‘proof of the pudding’ for orthologues has typically been functional complementation, although there are no reports that expression of human *LRPPRC* can functionally complement a *pet309* mutant. However, the function that the human and yeast proteins have in common, namely binding and stabilizing of *COX1* transcripts, is significant and, given the evolutionary distance between the two species, potential broader functionality of the protein in humans would not preclude there being orthologues.

#### 3.1.2. TACO1/DPC29

There are significant differences in structure between mtDNA-encoded transcripts in yeast and humans, meaning the vast majority of yeast mitochondrial translational activators do not have human homologues. In contrast to the majority of COX assembly factors, therefore, TACO1 is a mitochondrial translational activator that was first identified in mammals. Weraarpachai et al. described a patient with early-onset, slowly progressive Leigh syndrome resulting from an isolated COX deficiency, with a cytosine insertion (472insC) causing a frameshift in *TACO1* and a premature truncation of the protein [26]. In a very clear example of the challenge of identifying genotype–phenotype correlations, further reports of identical *TACO1* were associated with a broader spectrum of disease presentation, including ocular and cognitive impairments [40].

As with *LRPPRC*, *TACO1* was found to have a yeast homolog, *YGR021w*, with the translated proteins sharing only 29% identity at the amino acid level, but preliminary experiments in yeast did not reveal any translation defects and apparently wild-type levels of both growth on a non-fermentable carbon source and COX activity [26]. As is often the case, *YGR021w* was initially annotated during the sequencing of the yeast genome as encoding a protein of unknown function. In 2017, as part of defining the mitochondrial proteome, YGR021w was re-named DPC29 (delta-psi-dependent mitochondrial import and cleavage protein of ~29 kDa) [41]. There had been no further investigation of yeast DPC29, likely due to the lack of a readily discernible phenotype in the *dpc29* knock-out (∆DPC29), until a recent paper by Hubble and Henry that has significantly advanced our understanding of DPC29 [42]. These authors show that human TACO1 and *S. cerevisiae* DPC29 are predicted to have very similar structures and that both proteins associate peripherally with the inner mitochondrial membrane on the matrix side. Most critically, however, expression of human *TACO1* can functionally complement ∆DPC29 yeast, indicating that these proteins are indeed orthologs [42]. The experiments further suggest that DPC29 may act as a general mitochondrial translation factor and that it may function post-initiation, as mitoribosome profiling identified interactions with mRNA 3′-ends. Interestingly, the relationship of TACO1 and DPC29 mirrors that of LRPPRC and PET309, with one of the pair in each case being found to have a broader function in one of the species.

#### 3.1.3. C12ORF62/COX14

Mutations in *C12ORF62* have been reported for a single family in which the index patient suffered from a severe lactic acidosis that resulted in neonatal death. The mutation, which was identified through a combination of molecular genetic approaches, including microcell-mediated chromosome transfer, results in a M19I replacement [33]. Biochemical and cell biological analyses of this novel protein suggested a COX1-associated role in holoenzyme assembly, but the authors did not identify a connection to any of the known COX assembly factors.

Iterative orthology prediction through a program called Ortho-Profile, however, did identify *C12ORF62* as a divergent homologue of yeast *COX14* [28]. COX14, originally identified in yeast [29] and found to be associated with a high molecular weight complex, functions as a translational regulator of *COX1* that associates with SHY1 (surf homolog of yeast, discussed further below) and MSS51 [31]. Indeed, further studies of the molecular mechanisms of action for COX14 and MSS51 in yeast led to the discovery of COX25, another previously undescribed COX assembly factor that appears to function similarly to COX14 [43]. Only time—and further investigation—will tell whether either MSS51 or COX25 will eventually be found to have a human homolog and thereby potential involvement in inherited COX deficiencies.

#### 3.1.4. MITRAC12/COA3

As mentioned at the outset of this section, C12ORF62/COX14 and MITRAC12/COA3 function cooperatively to regulate the expression of *COX1* [44]. In contrast to the *C12ORF62* mutations described above, mutations associated with *COA3* were identified in an adult patient who presented with exercise intolerance and neuropathy, with a much milder clinical presentation more commonly associated with some mtDNA-based mutations [45]. In general, mutations affecting proteins that interact in a complex give rise to similar clinical phenotypes, but this case demonstrates the challenge in delineating genotype–phenotype correlations in COX deficiencies. The patient in this report was a compound heterozygote, with one allele encoding a truncated COA3 and the other generating a Y72C substitution in a conserved region of the transmembrane domain, resulting in the loss of COX1 and COX14 and an almost complete absence of assembled COX in fibroblasts [45].

COA3 was originally identified as CCDC56 in *Drosophila* but was also known as MITRAC12 through studies in HEK293 cells [27]. The connection between CCDC56/MITRAC12 and COA3 was, just as for C12ORF62 and COX14, identified through iterative orthology prediction [28]. COA3 has been extensively studied in yeast and was first identified through a genome-wide deletion screen [46] and found to encode an integral membrane protein that negatively regulates the expression of *COX1* [44]. COA3 was also shown to interact with COX14 to stabilize *COX1* intermediates [30] and to be a constituent of the MITRAC [27].

### 3.2. Defects Associated with Heme A Biosynthesis and Insertion

The catalytic core of COX requires the insertion of multiple prosthetic groups, including a high-spin heme A (*a*_3_) and a low-spin heme A (*a*), onto the nascent COX1 polypeptide, where the heme *a*_3_, together with Cu_B_, forms the oxygen-binding site of the enzyme. Heme A synthesis takes place in mitochondria and involves assembly factors COX10 [47,48] and COX15 [49,50,51], which function in a two-step process to convert protoheme (also known as heme B) to heme A. COX15 was additionally found to require the mitochondrial matrix protein, PET117 [52], which appears to be responsible for connecting the heme A biosynthetic pathway to the COX assembly pathway [53]. Ultimately, SURF1 (SHY1 in yeast) is believed to be the chaperone responsible for transferring heme A to the apoCOX1 during COX assembly [54,55,56].

#### 3.2.1. COX10

COX deficiencies resulting from mutations in *COX10* have been reported to be present in patients displaying a wide variety of different symptoms, including isolated COX deficiency; presentations varied from classical Leigh syndrome and anemia to fatal hypertrophic cardiomyopathy and sensorineural hearing loss [57,58]. The mutations described in the literature thus far document a combination of homozygous and compound heterozygous missense mutations, from both consanguineous and non-consanguineous pedigrees. Interestingly, a patient with a homozygous point mutation in the start codon presented with a Leigh-like disorder that proved fatal in infancy, as might be anticipated given a mutation that would effectively result in a *COX10* knock-out [59].

*COX10* was originally discovered in yeast and shown to encode a heme, A:farnesyltransferase, that catalyzes the conversion of heme B to heme O, which is the intermediate in the biosynthesis of heme A [47,49]. The human *COX10* orthologue was identified through a functional complementation screen of a human cDNA library in a yeast strain harboring a partial deletion of the *COX10* gene [48], directly identifying the human *COX10* as being orthologous to the yeast gene. This knowledge subsequently facilitated the direct corroboration of the negative impact of the mutations identified in patients on COX10 function. More recent studies in yeast identified another novel COX assembly factor, COA2, which stabilizes the COX10 complex and was identified through a yeast genetic suppressor screen [60,61]. Suppressor screens are an example of harnessing the power of yeast genetics, using mutant yeast strains that are either mutagenized or exposed to a selective pressure (i.e., forced to grow on a non-fermentable carbon source) to identify genetic changes that result in amelioration or changes to a mutant phenotype. Over the years, this powerful approach has identified many biologically relevant protein–protein interactions and improved our understanding of numerous fundamental cell biology pathways in both yeast and humans.

#### 3.2.2. COX15

Similar to the spectrum of different clinical phenotypes associated with mutations in *COX10*, COX-deficient patients bearing mutations in *COX15* also present with a wide variety of symptoms, resulting in cardiomyopathy or Leigh syndrome [62,63,64,65]. The patients comprised both homozygotes and compound heterozygotes, bearing a variety of missense mutations as well as a nonsense mutation that causes a premature truncation of the COX15 protein.

*COX15* was originally identified in yeast [50], and loss of COX15 was shown to result in lack/loss of heme A and increased levels of the heme O intermediate [51]. Because the human and yeast *COX15* are not orthologous, testing human mutations cannot use the functional complementation approach. However, an HPLC-based assay that was used to identify and quantify mitochondrial hemes in yeast mitochondria was adapted for heart and fibroblast mitochondria and used to demonstrate a decrease in heme A levels in a patient with *COX15* mutations [62]. There are significant challenges associated with studying compounds, like hemes A and O, that are extremely hydrophobic; likewise, both COX10 and COX15 are integral mitochondrial inner membrane proteins that are challenging with regard to expression and structural determination [66]. Early strides in understanding the heme A biosynthetic pathway were made through experiments in *E. coli* [67], which showed the *cyoE* gene is responsible for heme O synthesis. Further study of other bacterial cytochrome oxidases then expanded the pallet of known COX assembly factors [68], with *COX15* being homologous to *Bacillus subtilis CtaA* [51]. These homologies have lent themselves to heterologous expression of different *COX10* and *COX15* homologues in a number of different bacteria and have shown that the COX10 and COX15 proteins interact in a complex to achieve the synthesis of heme A [69]. Interestingly, an extension of the COA2 work mentioned above suggests that this small (<10 kDa) soluble mitochondrial matrix protein is also involved in the multimerization of both COX10 and COX15 [70]. Surprisingly, given that the COX15 homologues were proposed to use a monooxygenase reaction for heme A biosynthesis, it was found that the oxygen occupying the C8 formyl group was derived from water rather than molecular oxygen [71]. The challenges in working with highly hydrophobic compounds and proteins have precluded the elucidation of the precise mechanisms of action for both COX10 and COX15, but recent advances in structural modeling [66] should expedite future work in this direction.

#### 3.2.3. PET117

Mutations in *PET117* have only been reported in two siblings, from a second degree consanguineous family, both of whom were homozygous for a mutation that results in a premature truncation (termination of translation at position 58 of 81 codons) [72] of the PET117 protein. The siblings both presented with developmental delay and lesions of the medulla oblongata, with an isolated COX deficiency detected in both muscle and fibroblasts.

While originally identified in yeast several decades ago [52], PET117 has only recently become the subject of further investigation, likely because of its involvement in human disease. Taylor et al. have shown that yeast PET117 interacts with the heme A synthase and is necessary for the requisite oligomerization of COX15 [53]. The physical interaction of these two proteins further involves MSS51, a yeast-specific COX assembly factor that associates with COX14 but does not depend on SHY1 [53], which is discussed further below. A recent study using human cells suggests that PET117 stabilizes TACO1 through a direct interaction and thereby plays a role in regulating the expression of *COX1* [73]. Interestingly, PET117 has also been identified as interacting with MR-1S (myofibrillogenesis regulator 1), a human-specific COX assembly factor with no apparent yeast homologue, and PET100 [74], another COX assembly factor first identified in yeast [75]. This triumvirate of proteins associates to a greater extent with the nascent COX2 intermediate than with the COX1 module, suggesting a role in stabilization or coordination of the COX1 and COX2 assembly module intermediates [75].

#### 3.2.4. SURF1/SHY1

Mutations in *SURF1* were the first to be identified in association with any COX assembly factor [76]—and, in fact, in any nuclear gene encoding proteins associated with structure, function, or assembly of COX—in a series of patients with Leigh disease, otherwise known as subacute necrotizing encephalomyopathy [77], which is often accompanied by systemic COX deficiency. This discovery was made at a time when we did not yet have a complete human genome and provides an elegant example of combining cutting-edge technology with available yeast genetic information to zero in on a candidate gene. Using a combination of microcell-mediated chromosome transfer and gene mapping, Zhu and colleagues used a functional complementation approach to pinpoint the *SURF1* gene, whose yeast homologue, *SHY1*, had only recently been identified and characterized [54]. Knockouts of *SHY1* in *S. cerevisiae* result in a pronounced decrease in COX complexes, but a curious increase in cytochrome *c* content, as well as inability to grow on nonfermentable medium [54], while *SURF1* mutations in humans give rise to a COX-specific defect [76]. Indeed, mutations in *SURF1* appear to be the most common cause of the classical presentation of Leigh syndrome [78,79,80,81], although they have also been identified in a case of leukodystrophy [82], a mild encephalopathy without the typical MRI-identifiable lesions [83], and several cases of Charcot–Marie–Tooth disease [84]. There has been some characterization work carried out with the human SURF1 demonstrating that it is a mitochondrial membrane protein [85] and further suggesting involvement of SURF1 in facilitating the association of COX2 with the COX1 module [86] during assembly.

During this time, work in yeast has continued to provide further insights into the role of SHY1 in COX assembly, with early suggestions that the protein has a role in assembling the COX1 module [87], perhaps involving the Cu_B_-heme *a*_3_ center [88]. Modeling of the Leigh syndrome patient mutations in yeast demonstrated that SHY1 appears to have a role at the crucial intersection of COX assembly and regulation of COX1 synthesis [89]. Interestingly enough, the accepted role for SHY1, that of providing heme A to the nascent COX1 polypeptide, came from studies with prokaryotic oxidases, in which ablation of the *SURF1* homologue in *Rhodobacter spaeroides* resulted in about half of the COX complexes assembling incorrectly, as visualized by both mitochondrial cytochrome spectra and EPR analysis [55], which supported the yeast findings that SURF1/SHY1 may be required for assembly of the binuclear Cu_B_-heme *a*_3_ center. However, experiments involving heterologous expression of *Paracoccus denitrificans SURF1* homologs in *E. coli* show that SURF1 binds heme A, providing the most direct evidence to date that this protein delivers heme A to the assembling COX1 module [90].

### 3.3. Defects of Copper Acquisition at the Cu_B_ Site

In addition to two heme A molecules, the COX catalytic core contains a copper atom (Cu_B_) located in COX1, with the delivery of copper also requiring a series of chaperones and assembly factors. A full description of intracellular copper transport is far beyond the scope of this review, but the disposition of copper and copper-binding proteins found within mitochondria is relevant to our understanding of copper provision to the COX apo-enzyme. As depicted in Figure 2, copper is transported, via a series of transporters and chaperones, into the matrix via PIC2 and MRS3 [91,92], which provide the metal to the copper ligand, CuL, that has been identified as a source of copper in both the mitochondrial matrix and the cytoplasm [93,94]; CuL is an anionic, non-proteinaceous ligand that provides copper for mitochondrial cuproproteins. Through an as-yet unknown pathway, mitochondrial matrix copper is then exported to the intermembrane space for use by COX17 [95,96], which delivers the copper to either COX11 or SCO1, both of which reside in the inner mitochondrial membrane [96,97,98,99], with their functional domains residing on the intermembrane space side of the membrane. COX11 then transfers copper to the active site of COX1 [100,101], while SCO1 delivers its copper for the Cu_A_ site to COX2 [102] (discussed in Section 4). COX19 is a small COX17-like copper-binding protein [103,104] that interacts with COX11 in a redox-based manner and is essential for copper transfer [56,105]. PET191, which was also first identified in yeast [52], also appears to be involved in supporting the generation of the Cu_B_ site, possibly acting as a placeholder bound to COX11 prior to the delivery of copper by COX17 [56]. Mutations affecting copper trafficking to the Cu_B_ site of COX in patients were unknown until several recent reports of *COX11* mutations; to date, there have been no mutations identified in any of *COX17*, *COX19,* or *PET191*.

#### COX11

Mutations in *COX11* were first reported in 2022 in two unrelated patients—one with a homozygous missense (A244P) mutation that resulted in death within the first year, while a second patient had a milder disease course but was homozygous for a frame-shift mutation that results in a V12G substitution and a premature truncation of COX11 [106]. We recently reported the case of a patient with Leigh-like features who was compound heterozygous for a P247T substitution and T256Nfs*8, which results in a premature truncation in the C-terminal region of COX11 [107].

As COX11 was originally identified in yeast, much of the functional characterization of the COX11 protein has occurred in yeast. This can be attributed to the lack of amenable genome modification approaches in human cells, as well as the (relative) ease of working with yeast when *COX11* was first identified. As with most COX assembly mutants in yeast, the *cox11* null allele is characterized by a loss of the mtDNA-encoded subunits, mostly affecting COX1, a loss of the characteristic *aa*_3_ peak at 605 nm (detected through cytochrome spectral analysis), and retention of the nuclear-encoded subunits [97,108]. The protein was shown to bind copper in the Cu(I) state [109], and mutational analysis identified a number of essential residues, including the three conserved Cys residues and the amino acid residues found at the ends of β-strands and in the surface pocket behind the copper-binding loop [101]. Studies in yeast have also shown that loss of COX11 leads to a sensitivity to millimolar levels of exogenous hydrogen peroxide (H_2_O_2_) [101], although a copper transfer-competent COX11 is not needed, given that we identified several *cox11* mutants that were capable of partial COX assembly and yet were highly sensitive to peroxide and vice versa [110]. In spite of sustained efforts [110,111], we and others have not yet identified the specific role for COX11 in H_2_O_2_ metabolism.

## 4. Defects Affecting Synthesis and Assembly of COX2

The second module to form in the COX assembly pathway involves COX2, the mtDNA-encoded subunit that bears the binuclear copper site (Cu_A_) to which cytochrome *c* transfers electrons as part of the mitochondrial electron transport chain. The construction of the COX2 module begins with the co-translational insertion of COX2 into the mitochondrial membrane by OXA1L (OXA1 in yeast; oxidase assembly 1) [112,113,114,115], with the assistance of assembly factors COX16 [116,117], COX18 [118,119,120], and COX20 [121,122,123]. COX16 has a role in the recruitment of the metallochaperone proteins, the so-called SCO (synthesis of cytochrome oxidase) proteins, which transfer copper ions to the Cu_A_ site [99,117]; the copper transfer process also requires the thiol reductase COA6 [124,125,126]. Mammalian COX20 stabilizes the transient larger complex resulting from COX2 interacting with the metallochaperone complex [123], along with scaffold protein TMEM177 (which has no known yeast homologue) [127]. Although their roles in COX2 module assembly are less well understood, the process also involves assembly factors MR-1S, PET100, and PET117 [75]. Once the COX2 module is fully assembled with a completed Cu_A_ site, COX16 acts to bridge the COX2 and COX1 modules via interactions with MITRAC12 [117]. PET100 is also thought to act at this stage by stabilizing the combined COX1 and COX2 modules (S3 stage) prior to the addition of the COX3 module [75].

### 4.1. Defects Associated with COX2 Expression

#### 4.1.1. OXA1L/OXA1

There has been only a single report in the literature regarding *OXA1L* mutations underlying a mitochondrial disease presentation. The patient was a compound heterozygote with a nonsense mutation that generates a premature stop in the N-terminal half of the protein and a missense mutation that leads to an amino acid substitution (C207F) and exon skipping [32]. The patient presented with severe developmental delay and encephalopathy and was shown to have reduced assembly and activity of both COX and Complex III [32]. This finding was in contrast to an earlier study that used a knock-down approach in HEK293 cells to show that loss of OXA1L resulted in reduced complexes I and V, rather than COX (Complex IV) [128]. The contradictory results likely reflect not only the complexity of OXA1L function in human cells but also the inherent variability of different cell lines and tissues.

The majority of work on OXA1 has been carried out in yeast, in which *OXA1* was originally identified and shown to be required for respiratory competence [129]. Early studies of *oxa1* null mutants revealed an impairment of the processing and insertion of COX2 into the mitochondrial inner membrane [112,130], with OXA1 being required for the proper export of both the N- and C-termini of COX2. While initial observations suggested OXA1 was specific to COX assembly, further work in both yeast and humans has revealed a broader role for OXA1 in mitochondrial membrane insertion processes [12], including the import of the members of the mitochondrial metabolite carrier family of proteins [131]. Indeed, OXA1 has been shown to be a member of the YidC/Oxa1/Alb3 protein family, with roles in membranes from bacteria to thylakoids and mitochondria [132]. In the last decade, using a bioinformatics-driven approach, *OXA1* homologues were also identified in the endoplasmic reticulum, demonstrating the existence of an OXA1 superfamily whose members are involved in evolutionarily conserved membrane biogenesis processes [133,134]. With the very recent identification of a novel OXA1L-interacting protein, TMEM126A [135], the scope of OXA1/OXA1L actions in mitochondrial membrane protein biogenesis is becoming clearer. With increased use of technological advances, combining results from experiments in yeast and humans should lead to a complete understanding of the role of OXA1 in mitochondrial respiratory chain enzyme biogenesis.

#### 4.1.2. COX16

In a report of two unrelated patients, a homozygous nonsense mutation in *COX16* was found to underlie a clinical presentation of lactic acidosis with encephalopathy and hypertrophic cardiomyopathy [136]. Both patients had an isolated COX deficiency that was rescued through functional complementation with the wild-type *COX16*, using lentiviral transduction of fibroblasts.

*COX16* was first identified in *Saccharomyces cerevisiae* as encoding a small (118 amino acid residues) single-pass mitochondrial membrane protein [116]. While a human *COX16* homologue was identified, it did not functionally complement the yeast *cox16* knock-out strain and not much progress was made until recently, likely because no *COX16* mutations were identified in the intervening years [137]. A study in yeast had suggested an association for COX16 with both COX1 and the assembled COX holoenzyme [138], and these findings were then corroborated by further experiments with human COX16, which was found in association with both newly synthesized COX2 as well as COX1 assembly modules [117,139]. Interestingly, one of these studies also found that *COX16* knock-out cells retained significant COX activity [139], suggesting some level of redundancy with respect to COX16 function in human cells that is not the case in yeast cells. The overlap between the interactions within—and with—the COX1 and COX2 modules is increasingly being reported, and it seems likely that studies in yeast, with its lower degree of genomic and proteomic complexity, will be critical for further delineating the molecular mechanisms at play.

#### 4.1.3. COX18

A patient presenting with encephalo-cardiomyopathy in the neonatal period was reported to be homozygous for a *COX18* mutation that results in an D223H substitution, at a residue that is highly conserved, even in the distantly related OXA1L [140]; thus far, this is the only reported case of mutations in *COX18* underlying a mitochondrial disease phenotype.

*COX18* was originally identified in yeast as a mitochondrial membrane protein required to maintain steady-state levels of COX2 [118], with subsequent studies demonstrating that COX18 is responsible for exporting the C-terminal tail of COX2 across the mitochondrial inner membrane [119,141,142]. In keeping with the relationship to OXA1(L), yeast *COX18* expressed heterologously in *E. coli* can complement the Sec-independent function of YidC [143]. Not surprisingly, studies in human cells showed that COX18 functions as membrane insertase for nascent COX2 [120]. Given that the human COX18 homologue has a similar structure and subcellular localization to the yeast protein [144], results from studies in yeast will help to deepen our understanding of the molecular basis for disease arising from mutations in *COX18*.

#### 4.1.4. COX20

Mutations in *COX20* have been reported in more than 20 patients in a large number of different families from across the globe. In general, all the patients suffer from a neuropathy that varies in intensity, from mild to severe infantile forms [145]. The first report of mutations came from a patient with a moderate COX deficiency (about 40% of control levels in fibroblasts) who was homozygous for a mutation leading to a T52P substitution. The authors were able to show that COX assembly in the patient’s fibroblasts was blocked before the S3 stage, when the COX2 module should join with the COX1 module [146]. In the interim, there have been a number of different mutations identified, some of which are suggestive of founder effects [145,147,148].

*COX20* was identified in yeast [121], but the link to its human orthologue (*FAM36A*) was made through the use of the Ortho-Profile program, as the homology between *COX20* and *FAM36A* is not immediately obvious [28]. The yeast COX20 was shown to be a mitochondrial inner membrane protein, and the loss of COX20 resulted in the accumulation of the COX2 precursor [121]. Further experiments then verified a role for COX20 in processing of the COX2 leader peptide, export of the C-terminal tail of the polypeptide, as well as stabilization of COX2 by protecting it from proteolytic degradation [122]. In a good example of work in human cells building on work in yeast, a similar biochemical phenotype was observed in *COX20* knock-down human cells, with the additional observation that COX20 interacts with both SCO1 and SCO2, which are the metallochaperones for the Cu_A_ site on COX2 and are discussed in detail below. Interestingly, studies undertaken in yeast from an industrial bioethanol production perspective have recently found that *COX20* confers improved resistance to oxidative stress and apoptosis [149,150], which may have implications for the COX20 role in COX assembly.

#### 4.1.5. PET100

Several patients with severe lactic acidosis as a result of a COX deficiency, some presenting with Leigh syndrome, were found to harbor mutations in *PET100*. The first report presented eight patients, from six families, who were all homozygous for a null allele, with a mutation in the start codon [151]. A second mutation that results in a premature truncation of the protein was identified in a consanguineous family of different ethnic origin from that in the first report, demonstrating that *PET100* is also a potential mutational target in COX-deficient patients [152].

As with so many other COX assembly factors, *PET100* was first identified in yeast [74] as being required for COX assembly. PET100 was found to be associated with two different subassembly complexes, specifically one that contains the smallest nuclear-encoded subunits, COX7, COX8, and COX9, and another that contains subunits 5 and 6 [153], which are equivalent to COX4 and COX6 in the human COX. Further experiments should reveal the molecular mechanism(s) of action for PET100.

### 4.2. Defects in Copper Provision to the Cu_A_ Site

The SCO proteins are two sets of paralogues that arose independently in the yeast and human lineages [154], and consideration of the functions of the SCO proteins must be made separately. Nevertheless, as we will discuss below, human *SCO1* and *SCO2* mutations have successfully been modeled in yeast. The mitochondrial copper distribution network was described briefly in Section 3.2 (and Figure 2), with the delivery pathway to subunit 2 of COX in both yeast and humans involving the transfer of copper from COX17 [95] to the SCO proteins, which then transfer the copper to the Cu_A_ site. In yeast, copper is transferred to SCO1 [96,102,155,156], which then transfers Cu(I) to the Cu_A_ site through either direct or indirect means [157,158,159,160]. In humans, SCO1 (hSCO1, for the purpose of this review) and SCO2 (hSCO2) have differentiated, non-overlapping roles in the transfer of copper from COX17 to the Cu_A_ site [158], with the two SCO proteins forming a ternary complex with apo-COX2 and each subsequently transferring a single copper ion to COX2 [161]. In both yeast and humans, the copper transfer process also involves the thiol reductase COA6 [162], which reduces disulfide bonds in both SCO1 and SCO2 to keep them functional in copper transfer to COX2 [125,126].

#### 4.2.1. hSCO1

*hSCO1* mutations have not been reported in the numbers seen in genes encoding some other COX assembly factors, such as *SURF1* and *SCO2* [163]. The first report came from a neonate with hepatic failure and lactic acidosis and a COX deficiency identified in both muscle and liver [164]. The patient was a compound heterozygote, with a small deletion leading to premature truncation on one allele and a missense mutation leading to a P174L substitution (adjacent to the copper-binding domain) on the other allele [164]. A second patient with hypertrophic cardiomyopathy and a homozygous G132S mutation, who died in the neonatal period, was also reported [165]. Interestingly, another *SCO1* missense mutation, which leads to a M294V substitution, was identified in a case of encephalopathy that was not as severe as those described previously and led the authors to propose a genotype–phenotype correlation with respect to *SCO1* mutations [166].

*SCO1* was originally discovered in yeast through its respiratory-deficient phenotype, typical of a *pet* strain [98,167], and found to function as a high copy suppressor of a mitochondrial copper recruitment defect: the respiratory competence of a ∆COX17 strain could be restored by overexpression of *SCO1* [99]. COX17 was suggested to function upstream of SCO1, as overexpression of *COX17* was not able to rescue the ∆SCO1 mutant, while overexpression of *SCO1* could rescue the *cox17* null mutant. The SCO1 protein was therefore proposed to shuttle copper from COX17 to COX2 [102]. Indeed, mutations in the CxxxC motif, which was proposed to bind copper, render SCO1 incapable of supporting COX assembly, resulting in respiration-deficient cells [157,168]. Further studies of yeast SCO1 demonstrated that the protein binds Cu(I) [155] and has a structure similar to that of hSCO1 [169]. Like COX11, SCO1 is a transmembrane protein with a short N-terminal tail in the matrix and the bulk of the protein located on the intermembrane space side of the mitochondrial inner membrane. Interestingly, SCO1 was proposed to have a secondary role, in addition to that in copper transport, as structural analysis revealed a thioredoxin-like fold, suggesting a possible redox activity for the protein [169,170]. In another similarity between COX11 and SCO1, a *sco1* null mutant was shown to be sensitive to millimolar quantities of exogenous H_2_O_2_ [170], suggesting a role in metabolizing peroxide. The potential roles of SCO1 and COX11 in metabolizing peroxide are thought to be distinct from their roles in COX assembly and respiration, as some respiration-deficient *sco1* mutants were able to resist peroxide to a greater degree than the ∆SCO1 mutant; likewise, some *sco1* mutants displayed peroxide sensitivity but not a respiratory deficiency [110]. Interestingly, while the disconnect between respiratory function and peroxide sensitivity has been supported by others [171], there are conflicting reports as to whether or not the copper-binding ability of SCO1 is required for its peroxide sensitivity [110,171], a result that could be due to differences in strain background. There have been few *SCO1* mutations reported in human disease, but the P174L mutation was studied in yeast and shown to have defective copper transfer from COX17 but normal copper-binding activity [172].

#### 4.2.2. hSCO2

*SCO2* mutations in human disease have been found in association with a wide variety of clinical presentations. Mutations in *hSCO2* were first identified in a series of unrelated infants presenting with fatal cardioencephalomyopathy resulting from a COX deficiency [154]. All three probands were compound heterozygous, all bearing the missense variant E140K; two of the patients further had a nonsense mutation that resulted in a premature truncation (Gln52*), while the other patient bore a second missense mutation leading to a S225F substitution [154]. Compound heterozygotes with COX deficiencies and *SCO2* mutations were also identified in several patients with lethal infantile cardioencephalomyopathy, each of whom had the E140K substitution [173], which has also been found in other patients by different research groups [1]. Interestingly, there have been several reports of *hSCO2* mutations associated with spinal muscular atrophy presentations, both involving compound heterozygous patients with the widely documented E140K mutation. In one case, the second allele contained a nonsense mutation resulting in a premature truncation (Trp36*) [174] and, in another report, the mutation on the second allele resulted in a C133Y substitution in the copper-binding site [175].

As described above, the two SCO proteins in yeast are paralogues of the human SCO proteins, hSCO1 and hSCO2, which precludes characterizing human *SCO1* and *SCO2* mutations directly through functional complementation in ∆SCO1 yeast. Nonetheless, the functional consequences of human mutations have been studied by generating the homologous mutations in yeast *SCO1*, which has proven to be a fruitful approach. The first *hSCO2* mutations were successfully modeled in yeast and led to several novel insights. The E155K (E140K in humans) mutation did not result in a respiratory deficiency in yeast, suggesting it might be a hypomorphic allele in compound heterozygous patients. This was corroborated in a subsequent report in which a set of patients were found to be homozygous for the E140K mutation and had a relatively later onset of clinical symptoms [176]. The analysis of the yeast S240F (S225F in humans) mutation identified a blue shift in the mitochondrial cytochrome spectrum, indicating that heme A was in an altered environment, and showed that the mutant had almost wild-type levels of COX1 but no detectable COX2 [102]. These results provided the first suggestion that SCO1 might be providing copper strictly to the Cu_A_ site on COX2, which has since been corroborated using multiple approaches [177,178].

Unlike its human homologue, yeast *SCO2* is not required for COX assembly, although overexpression of *SCO2* was found to partially complement a yeast *SCO1* point mutant [99,179]. While SCO2 and SCO1 in yeast share approximately 50% identity and are both ~30 kDa integral components of the mitochondrial inner membrane, with a very similar topology, to date, there has been no phenotype found in association with the *sco2* null mutant, leaving the function for SCO2 in yeast unknown.

#### 4.2.3. COA6

Through a next-generation sequencing-based approach, a single compound heterozygote bearing mutations in *COA6* was identified, with a combination of a missense mutation and a nonsense mutation resulting in hypertrophic cardiomyopathy in the affected individual [124,180].

COA6 was originally identified in yeast, not in the ‘classical’ approach with a complementation group in a mutant collection, but rather through the application of a proteomics-based approach [162] that defined the proteome of the mitochondrial intermembrane space. The authors then demonstrated a specific reduction in steady-state levels of both COX2 and COX3 in a *coa6* null mutant, with the expected reduced growth on non-fermentable carbon sources [162]. The fact that the respiration deficiency could be rescued by copper supplementation further supports the involvement of COA6 in metalation of the Cu_A_ site [124]. Further analysis showed that COA6 interacts with COX2, the SCO proteins, and COX12, which is the yeast homologue of nuclear-encoded subunit 6B in human COX [181]. The human *COA6* was also reported to be orthologous with the yeast *COA6*, although the functional complementation required the use of a hybrid construct that contained the N-terminal portion of the yeast protein and the C-terminal two-thirds of the human protein [181].

## 5. ‘Other’ COX Assembly Factors

In this review, we have highlighted 18 proteins with a demonstrated role in the COX assembly process, but there remain a number of COX assembly factors with poorly defined or unspecified roles in the various or combined assembly pathways, some of which have also been found to underlie inherited COX deficiencies. One such factor is COA5 (encoded by *C2ORF64*), which was identified by the iterative orthology approach that has successfully identified the human homologues for a number of other COX assembly factors [28]. Two siblings presenting with a fatal neonatal cardiomyopathy were found to have a severe COX deficiency in heart muscle and fibroblasts and to be homozygous for an A53P mutation [182]. Preliminary analyses of patient fibroblasts further suggested that COX assembly was negatively affected at an early stage in the COX assembly pathway. The yeast homologue of *COA5* is *PET191*, which was isolated and identified from a large COX mutant collection [52]. Biochemical and genetic characterization of PET191 suggests it exists in a large oligomeric complex but that, unlike other twin Cx_9_C motif proteins, the import of PET191 into the intermembrane space is not reliant on the MIA40 import pathway [183]. There have been no other reports regarding PET191, and its role in COX assembly remains largely uncharacterized.

In a similar vein, there are a number of well-characterized yeast COX assembly factors with human homologues or orthologues for which mutations have not (yet) been identified in association with human mitochondrial disease. Some of these, such as *COX17* [95] and *COX19* [103], were identified more than 20 years ago from mutant collections and have thus been widely investigated as candidate genes in patients with COX deficiency of unknown etiology. Others, such as *COA4*, have been identified more recently using a number of different genetic and proteomic approaches. COA4 was originally identified as CMC3 [184] in a proteomic screen for twin Cx_9_C motif proteins found in the mitochondrial intermembrane space. Interestingly, *COA4* was also isolated as a multi-copy suppressor of a *shy1* point mutant [185]. Characterization of the *coa4* null mutant suggested a role for this protein at a step following the assembly of S1, the COX1-containing module [185]; a more recent study has identified *COX11* as a multi-copy suppressor of a *coa4* null mutant [186], suggesting a role for COA4 in copper transfer to the Cu_B_ site. Clearly, further study is required to clarify the role of COA4 in COX assembly; the available data support the inclusion of human *COA4* as a candidate gene in COX-deficient patients.

While we note that mutations have also been identified in cases of leukoencephalopathy in both *COA7* [187] and *COA8/APOPT1* [188,189], neither of these human COX assembly factors appear to have yeast homologues and were therefore not discussed in any depth. Through a variety of different approaches, the identification of new COX assembly factors, as well as homologues and orthologues, in both yeast and humans is ongoing, and orthologous proteins for these two COX assembly factors may well yet be uncovered.

## 6. A Future for Yeast in Studying Human COX Defects

For all the significant advances that have been achieved in improving our understanding of COX assembly in health and disease, challenges and contradictions remain. As just one example, we still do not fully understand the molecular underpinnings of heme A biosynthesis and addition to COX1, strongly suggesting we will identify further players in this essential part of the COX assembly process. The continued use of large-scale screens, combined with targeted experimentation in both yeast and human cells, will undoubtedly contribute to improving our understanding of the molecular pathways that underpin COX assembly. A brief scan of information in several publicly accessible databases reveals that many of the currently recognized COX assembly factors participate in a myriad of protein–protein interactions and suggest there are further proteins involved in aspects of COX assembly that are yet to be discovered. In addition, the presence of COX assembly factors in interaction networks further supports the possibility of additional roles for these proteins in other cellular functions and pathways, so called ‘moonlighting’, a concept that is proving to be much more widely distributed than perhaps initially thought [190,191]. COX11 and SCO1, which have well-understood roles in copper transfer to the subunits of the COX catalytic core, also have an as-yet poorly understood role in peroxide metabolism and thereby serve as a relevant example of proteins engaged in ‘moonlighting’ activity.

As is evident from the descriptions of the clinical cases of COX deficiency we have discussed, many of the patients are compound heterozygotes. An unresolved question is that of hypomorphic alleles in these uniformly autosomal recessive disorders, especially in those cases in which studies in haploid yeast have found that only one of two alleles gives rise to a respiration-deficient phenotype [102,107]. A dominant-negative phenotype has been described for overexpression of *hSCO1* and *hSCO2* [158], but how this relates to the clinical outcomes for compound heterozygous patients remains unclear and requires further study.

In addition to the importance of COX assembly factors in human mitochondrial disease, there is an increasing interest in mitochondrial proteins, especially those with redox functions, such as the twin Cx_9_C motif proteins, in the broader spectrum of human disease, especially cancers and neurodegenerative diseases [21]. While involvement in the disease presentations of multiple neurodegenerative diseases, such as Parkinson and ALS, has long been known, the relevance of mitochondrial proteins to cancer is a more recent phenomenon. While this is not surprising in light of the Warburg effect [192], the realization that mitochondrial metabolism may serve as a treatment target is gaining traction, and there have been numerous reports describing the involvement of COX assembly factors, including *COX16* [193] and *COA4* [194], in breast cancer and *COA1* [195] in bone cancer. *hSCO2* has long been known to be regulated by p53 [196], and down-regulation of COX17 has been proposed as a means to inhibit metastases in triple-negative breast cancer [197]. Indeed, loss of COX17 has been reported to impair DNA methylation and self-renewal of leukemic stem cells in acute myeloid leukemia [198]. Clearly, more precise information regarding the functions of individual COX assembly factors, whether in yeast or humans, will benefit our understanding of diseases, such as cancers, beyond the mitochondrial disease arena.

For all the reasons enunciated at the outset, yeast is a tractable and fruitful model system. In addition, the reduced complexity of yeast in terms of total numbers of genes and proteins, along with the well-developed techniques in yeast genetics and molecular and cellular biology that have stood the test of time, have allowed us to understand the intricacies associated with mutations affecting COX assembly. Another advantage of yeast is that this organism can help us cut through the confusion created by our inability to draw genotype–phenotype conclusions for most human mitochondrial diseases. Table 2 presents a summary of clinical phenotypes associated with the COX deficiencies we have described here, along with the very wide range of symptoms and systems impacted by those mutations. When we consider that this constellation of clinical presentations arises through mutations in a small subset of proteins dedicated to just one cellular pathway, namely that of assembling the COX holoenzyme, the challenge of drawing genotype–phenotype correlations for these devastating diseases is immense, and working within a simplified model system has enormous advantages, especially given that COX defects result in anomalies in cellular redox homeostasis that are replicated in yeast. The advantages of the yeast model system also extend to broader mitochondrial defects affecting the other components of the oxidative phosphorylation machinery, as well as mitochondrial protein import, proteostasis, mitophagy, and oxidative stress responses [199].

One of the most powerful examples of how studies in yeast have [200] and continue to have [201] a central role in understanding human disease is the identification of the building blocks of the cell cycle. Yeast clearly has a rich history as a model system for better understanding a wide variety of cell biological processes in higher eukaryotes, including for the inherited COX deficiencies described in this review. It is clear that basic discoveries in yeast have informed and facilitated the identification of mutations in patients with defects in COX assembly. There are still no ‘proven’ therapies for the vast majority of COX deficiencies and, given the ‘collateral damage’ that defects in oxidative phosphorylation inflict on cellular function and homeostasis, the development of therapies is highly sought after—but also a daunting task [202]. As seen over the history of science and medicine, a better understanding of disease processes can lead to the development of appropriately targeted (and thereby improved) treatment modalities. There is clearly still a future for studies in yeast that will serve to complement experiments conducted with human cells and tissues and thereby continue to make critical contributions to our understanding of inherited COX deficiencies.

## Figures and Tables

**Figure 1 ijms-25-03814-f001:**
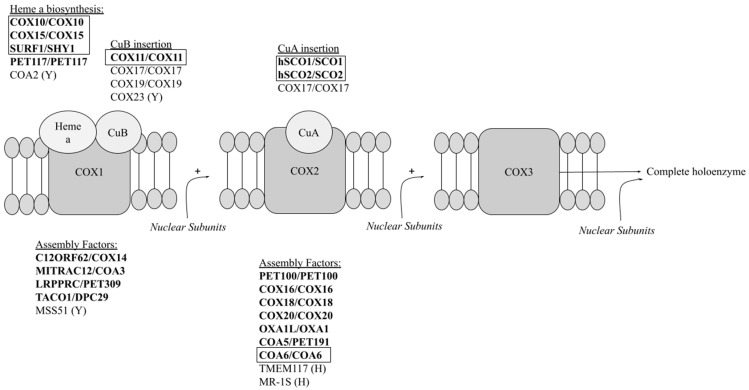
Proteins involved in COX assembly in yeast and humans. The schematic illustrates the level at which the assembly factors discussed in this review are involved, i.e., COX1 module, focusing on the assembly factors discussed in this review. The placement of COA5/PET191 is arbitrary, as its specific role has not yet been delineated. The essential prosthetic groups (heme A, Cu) found on subunits 1 and 2 are indicated in spheres. Mutations in genes encoding proteins (human protein/yeast protein) labeled in bold have been shown to cause inherited COX deficiencies; mutations in human genes encoding bolded proteins in boxes have been studied directly in yeast. (Y) = found only in yeast; (H) = found only in humans.

**Figure 2 ijms-25-03814-f002:**
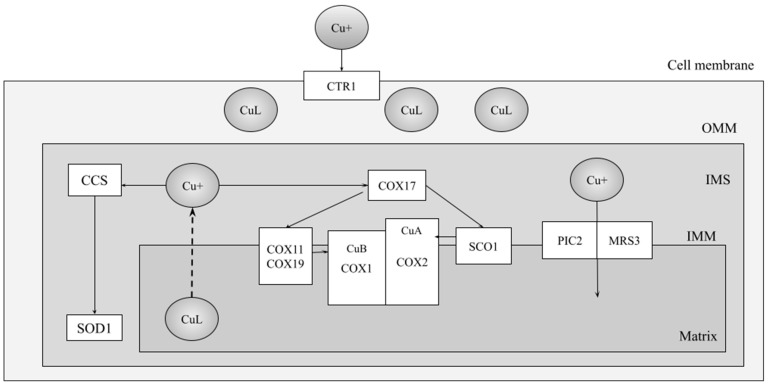
Copper trafficking within the mitochondrion. The schematic depicts the movement of copper from the extracellular space, via CTR1, and to the mitochondrial matrix via PIC2 and MRS3. COX11 and COX19 are depicted in a single box because they have been shown to act together to transfer copper to COX1. The two major cupro-proteins acquiring copper from the matrix CuL are COX and SOD1, the copper-zinc superoxide dismutase, with CCS serving as the copper chaperone for SOD1. The hatched arrow (----) indicates that the mechanism for copper export to the IMS remains unknown. We have shown the yeast proteins and pathways and not included other copper transporters and chaperones, as they are not as directly relevant to inherited defects of the COX assembly pathway and hence, to human COX deficiencies. OMM = outer mitochondrial membrane, IMS = intermembrane space, IMM = inner mitochondrial membrane, CuL = copper ligand.

**Table 1 ijms-25-03814-t001:** Overview of human disease-associated COX assembly factors and their yeast homologues.

Human Protein	Yeast Homologue	Role(s)
*COX1 Module-Associated*		
LRPPRC	PET309	COX1 mRNA stabilization, activation of transcription
TACO1	DPC29	Translational activator for COX1, other mtDNA transcripts
C12ORF62	COX14	Regulates COX1 expression, part of MITRAC
MITRAC12	COA3	Regulates translation of COX1; modulates binding to COX2 module via COX16
COX10	COX10	Farnesyl transferase (heme O synthase)—converts heme B to heme O
COX15	COX15	Heme A synthase—converts heme O intermediate to heme A
PET117	PET117	Required for oligomerization of COX15, hemylation of COX1
SURF1	SHY1	Involved in the final hemylation of COX1
COX11	COX11	Delivers copper to COX1
*COX2 Module-Associated*		
OXA1L	OXA1	Insertion of mitochondrially encoded subunits into IMM
COX16	COX16	Chaperone for COX2, recruits SCO proteins; helps COX2 module associate with S2; brings COX1 and COX2 modules together
COX18	COX18	Insertion of the C-terminus of COX2 in the IMM
COX20	COX20	Binds to COX2 before and after cleavage; stabilizes complex with SCO proteins
PET100	PET100	Interacts with MR-1S, PET117 in late stages of biogenesis; essential to assembly in humans; stabilizes S3 intermediate
hSCO1	SCO1	Insertion of copper into Cu_A_ site; hSCO1 associates with PET191 prior to copper delivery by COX17, passes one Cu to COX2.
hSCO2	SCO1	hSCO2 undergoes disulfide exchange with COX2 and delivers Cu; yeast SCO2 function unknown
COA6	COA6	Thiol reductase activity, Cu_A_ site assembly; perhaps overlapping role with hSCO2
*Unspecified Role*		
COA5	PET191	Essential to human assembly; associates with SCO1 until Cu is delivered

**Table 2 ijms-25-03814-t002:** Clinical phenotypes for human COX assembly factor deficiencies.

Assembly Factor	Phenotype	Citations
*Factors Associated with COX1 Module*		
LRPPRC	French-Canadian Leigh syndrome	[23,37]
TACO1	Leigh syndrome, ocular and cognitive impairments	[26,40]
COX14	Fatal neonatal lactic acidosis	[33]
COA3	Obesity, exercise intolerance, short stature, neuropathy	[45]
COX10	Tubulopathy and leukodystrophy, Leigh syndrome and fatal infantile hypertrophic cardiomyopathy, sensorineural hearing loss	[57,58,59]
COX15	Fatal infantile hypertrophic cardiomyopathy, Leigh syndrome	[62,63,64,65]
PET117	Neurodevelopmental regression, medulla oblongata lesions	[72]
SURF1	Leigh syndrome, leukodystrophy, mild encephalopathy, Charcot–Marie–Tooth disease	[76,77,82,83,84]
COX11	Infantile-onset mitochondrial encephalopathy, Leigh-like features	[106,107]
*Factors Associated with COX2 Module*		
OXA1L	Mitochondrial encephalopathy and combined oxidative phosphorylation defect	[32]
COX16	Hypertrophic cardiomyopathy, encephalopathy and severe fatal lactic acidosis, liver dysfunction	[136]
COX18	Neonatal mitochondrial cardioencephalomyopathy and axonal sensory neuropathy	[140]
COX20	Early-onset hypotonia, ataxia, areflexia, dystonia, dysarthria, and sensory neuropathy	[145,146]
PET100	Leigh syndrome, Infantile lactic acidosis	[151,152]
SCO1	Neonatal-onset hepatic failure and encephalopathy, hypertrophic cardiomyopathy	[164,165,166]
SCO2	Fatal infantile cardioencephalomyopathy, hypertrophic cardiomyopathy, spinal muscular atrophy	[154,173,174,175,176]
COA6	Neonatal hypertrophic cardiomyopathy	[180]
*Unspecified Role*		
COA5	Fatal infantile cardioencephalomyopathy	[182]

## Data Availability

Not applicable.

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
