# Peer review of "More than Just Bread and Wine: Using Yeast to Understand Inherited Cytochrome Oxidase Deficiencies in Humans"

_ijms, 2024, doi:10.3390/ijms25073814_

Round 1

Reviewer 1 Report

Comments and Suggestions for Authors

In the present manuscript, Chenelle A. Caron-Godon et al. highlight the advantages of the yeast Saccharomyce cerevisiae for the study of Cytochrome Oxidase, listing all the examples where yeast has been very useful for investigating the role of proteins involved in complex IV biogenesis. Yeast is indeed a very useful organism for the study of mitochondria and mitochondrial diseases in general. It is very useful to summarize what is known specifically about complex IV. Brischigliaro and Zeviani's 2001 review in BBA also focuses on human diseases linked to complex IV deficiencies, also mentioning the existence of a yeast model where one exists, but in this review the authors go further, explaining what yeast has contributed to our understanding of the role of each protein and what can yeast continue to contribute in this respect. This summary has not been done recently.

The review is well-structured, comprehensive on the subject and very well documented with recent references.

Minor points :

Please replace DPC1 by DPC29 in line 224.

To be consistent, please add MITRAC12/COA3 in line 244

Please replace rpesented by presented in line 336.

Please replace HEK239 by HEK293 in line 467.

Please replace it’s by its in line 689.

Reviewer 2 Report

Comments and Suggestions for Authors

In the manuscript authors describe roles of many proteins required for functioning of cytochrome oxidase and how mutations in genes encoding these proteins translate to  cytochrome oxidase deficiencies and human genetic diseases. The review is comprehensive and generally well written, however, requires some corrections.

General comments

The convention was used that both yeast and human protein names are written in uppercase letters. Note about this is on P 19 L785-787. This is too late . This note should be given earlier, as first yeast proteins appear or in the legend of Figure 1. Link to the convention of nomenclature should be given explaining who and when recommended it.

Gene names are in italics to distinguish from  names of encoded proteins. Authors have to be careful and write the name in italics, or not, depending if the sentence is about genes or proteins. Better do not mix genes and proteins in one sentence to be clear.

Mutations are in genes. Genes are expressed, not proteins. Proteins are synthesized or produced. All other statements are laboratory jargon.

In the last chapter I am missing some general information what are the cellular effects of COX deficiency,  defects of oxidative phosphorylation, defects in mitochondrial protein transport, proteotoxic stress, oxidative stress?. All these kind of effects can be studied deeply using yeast.

Specific comments

P3, L99 carefully coordinated action

P 3 L123 COX deficiency-associated mutations

P3, L128, COX assembly proteins have been found to be defective in cases

P3, L129  mutations in genes encoding assembly factors

P3, L130  nuclear genes encoding COX subunits

P4 Figure 1  L140 mutations in genes encoding proteins

P4, L141 mutations in human genes encoding bolded proteins

P4, L154 Translational regulation of human COX1 synthesis (or COX1 expression)

P6, L178 including COX3 mRNA (including COX3 transcript)

P6, L187 PET309 and COX1 transcripts

P6, L190 expression of human LRPPCR can functionally complement pet309 mutation. Complementation is a genetic term and concerns genes, not proteins.

P6, L192 COX1 transcript

P6, L208 at the codon level. The gene is not built of amino acids.

P7, L237 translational regulator of COX1 mRNA

P7, L246 expression of COX1.

P7, L256 as CCDC56 in Drosophila. (the sentence is about proteins)

P7, L259-261 COA3…. was first identified through a genome-wide deletion screen….. the expression of COX1.

P7, L261 COA3 protein was also shown to interact with COX14

P7, L275 reported in 2000 to be present in patients (reported in a paper)

P9, L335 (termination of translation at 58 of 81 codons). Codons are translated into amino acid residues in a protein.

P9, L345 expression of COX1. (or synthesis of COX1?)

P9, L353-354 any gene encoding COX assembly factor – and, in fact, in any nuclear gene encoding protein associated with structure, function or assembly of COX

P10, L 384 SURF1 homologs in E. coli document/show that (experiment is not a person)

P10, L393-394 is not depicted. MRS3 is not in Figure 2. Generally the Figure 2 does not corresponds well with the text in 3.3. COX19 is also not in the picture.

P11 Figure 2 What is Ccs? It is not in the legend and not in the text 3.3. Are human or yeast proteins shown in the picture of Figure 2?. It is not written in the title or legend. In yeast we also have Ctr2 copper transporter.  Make it clear.

P12, L434 amino acid residues

P12, L462 nonsense mutation that generates a premature stop codon. (stop codon does not encode anything)

P12, L473 processing and insertion of COX2 into  the … membrane

P12, L475-476 OXA1 (Oxidase Assembly 1) should be explained earlier, L446, when OXA1 first appeared.

P13, L496 amino acid residues

P13, L518 do not start a sentence with AND.

P13, L519 COX18 expressed heterologously

P14, L568 in Section 3.2 and Figure 2

P14, L580 hSCO1 mutations… seen in genes encoding some other COX assembly factors

P15, L587 SCO1 missense mutation

P15, ?L590 SCO1 mutations.

P15, L593  ΔCOX17 strain

P15, L595 ΔSCO1 mutant. Why in L609 is sco1 mutants  but ΔSCO1 here and in L610? Not consistent nomenclature. Why not sco1 mutants and sco1Δ mutants (small letters) everywhere?

P16, L636 why not sco1Δ yeast?

P16, L651 sco1 point mutant? I do not see the logic in yeast genetic nomenclature.

P18, L746 gaining attraction?

P18, L765 mutations in a small subset of genes encoding proteins dedicated
